# Temporal Movement of a Dieback Front in a Population of Parkinsonia in Northern Australia

**Naomi D. Diplock** [1,2] **and Victor J. Galea** [1,*]

1   School of Agriculture & Food Sciences, Gatton Campus, The University of Queensland,
    Gatton, QLD 4343, Australia; naomi.diplock@uqconnect.edu.au
2   Applied Horticultural Research, Suite 340/1 Central Ave., Eveleigh, NSW 2015, Australia
*   Correspondence: v.galea@uq.edu.au; Tel.: +61-7-5460-1282

**Abstract:** The temporal progress of *Parkinsonia aculeata* dieback through a well-established, naturally occurring dieback affected site was monitored using two transects over a seven-year period. This revealed the time and spatial dynamics underlying the nature of this disorder. Assessment of this site demonstrated a decline in individual plant health over consecutive years, with 98% of parkinsonia plants dying over the study period. Minimal recruitment of new plants led to a collapse in the parkinsonia population. *Macrophomina phaseolina* (*Botryosphaeriaceae*) was the only species with known pathogenicity on parkinsonia found in the transect site. This information provides a valuable insight into the timeframe involved in this disease process from infection through to plant death. This is the first research to date to assess the temporal movement of parkinsonia dieback.

**Keywords:** *Parkinsonia aculeata*; *Botryosphaeriaceae*; temporal movement; dieback; *Macrophomina phaseolina*

## 1. Introduction

Parkinsonia (*Parkinsonia aculeata* L. (Fabaceae)) is an introduced and invasive thorny shrub/tree considered to be a significant weed problem in the rangelands and natural riparian environments of northern Australia [1]. A disorder killing parkinsonia plants has been observed in Hughenden, Queensland, Australia since the 1950s with no other species in the area showing similar dieback symptoms (Tony Kendall, 2006, personal communication). More recently, this same phenomenon has been observed in numerous sites across northern Australia and is commonly referred to as parkinsonia dieback [1–4]. Anecdotal evidence suggests that in some regions across Australia, dieback may be preventing parkinsonia from becoming a greater problem [2]. Dieback (otherwise described as blight or decline) is a term often used when the cause is unknown, and is used as a descriptor of a disease symptom [5]. High, unexplained mortality has been observed among adult and juvenile parkinsonia trees from individual plants to entire stands. In most cases, parkinsonia dieback can be recognised by symptoms starting from the tip of the branch, which begins to die back, with phyllodes and pinnae drying up and remaining attached to the plant. A distinct dark, necrotic region on the stem appears to move down the plant as the disorder develops preceding plant death [6] (Figure 1). Affected stems are killed throughout with all tissues, pith, vascular elements, and bark being fully dehydrated and brittle. Often, affected plants are soon colonised by stem boring insects and termites.

Understanding the temporal behaviour of dieback, and the role it plays in the population dynamics of parkinsonia is crucial from a management perspective, to determine both the role that dieback can have in control strategies and to provide an insight into the timeframe involved in this disease process from infection through to plant death. This is the first research to date to assess the temporal movement of parkinsonia dieback.

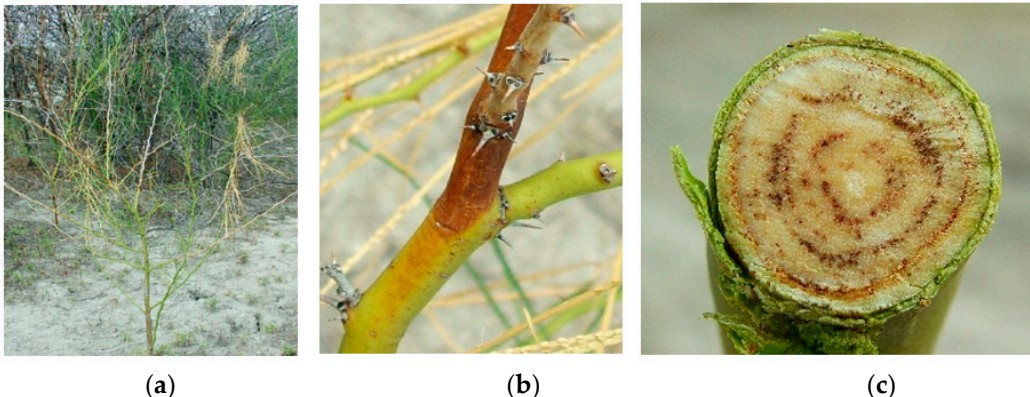

(**a**)          (**b**)          (**c**)

**Figure 1.** Typical symptoms of dieback on *Parkinsonia aculeata*. Dieback affected plant (**a**); Dieback lesion moving from tip to a position lower on the branch (**b**); Stem vascular bundle staining (**c**).

Transect sites were established in Hughenden, QLD, Australia to monitor the progress of parkinsonia dieback through a well-established, naturally occurring dieback affected site. At this location, minimal control has been required as parkinsonia trees are left to naturally succumb. Observations nearby suggest that dieback is capable of completely controlling parkinsonia populations, with very low seedling recruitment observed. Where new plants have emerged, management efforts were not focused on these plants, as they appeared to be serving as a host for the (fungal) causal agent of dieback, possibly preventing any new major infestations (Tony Kendall, 2006, personal communication). However, other properties in the same region have not experienced dieback naturally in parkinsonia, and control is necessary (Tony Kendall, 2006, personal communication).

The behaviour and movement of plant diseases in stands of trees has largely been understudied at the landscape level [7,8]. The impact of pathogens and their effect on landscape structure and composition is limited to the selection of individuals which are affected being less vigorous, or genetically unfit to withstand infection [7,9], with the interaction of biotic and abiotic factors influencing which individuals succumb to disease [10,11]. If a particular plant species within a plant community is susceptible to attack, pathogens could potentially control the occurrence of that particular species, especially when plants are killed before reproducing [12].

Transects were established primarily to monitor the progress of dieback over an extended period in a site where it naturally occurs in a parkinsonia population. Furthermore, these transects would establish a general base line understanding of the time taken for an adult parkinsonia tree to be killed by dieback after the first observed signs of infection. Observations of seedling recruitment in an area with an assumed high inoculum load were also investigated at these transects. Other sites on this property were known to have been completely cleared of parkinsonia because of naturally occurring dieback, indicating that environmental conditions were favourable for dieback in this location. It was anticipated that this site would provide suitable conditions for monitoring dieback movement through the parkinsonia stand in a relatively short timeframe. Another advantage of this site was a clear dieback front moving through a stand to healthy parkinsonia which was sparse enough to allow access for monitoring.

This paper presents the findings of a seven-year study of a naturally occurring parkinsonia dieback site. It was hypothesised that the health of individual plants would decline over time eventually leading to death, with minimal recruitment due to infection by dieback pathogens. The aim of this study was to assess over time, the health of parkinsonia plants along two 50 m × 8 m transects to develop a spatial disease model.

## 2. Materials and Methods

Two transects were established in June 2005 on a flood plain of the Flinders river at Koon Kool Station (lat 20°40′37″; long 144°20′16″), a cattle property north east of

Hughenden north QLD, a semi-arid area, receiving a typical annual rainfall of 450 mm [13]. Water movement along this flood plain is in an SSW direction with elevation of the transects dropping approximately 1 m over the 50 m length. Hughenden experiences summers with a mean maximum temperature of 37 °C in December and cool winters and a mean minimum temperature of 9 °C in July [14].

The 50 m long and 8 m wide transects were established approximately 2 km from the nearest creek. The dieback front appeared to move along the flood plain. The position of these transects were chosen to include both dieback-affected plants and healthy plants. The origin point of dieback was considered as the single dead plant present in each transect at the time of trial establishment. Transect 1 running at approximately 45° to the dieback front had plants of its northernmost end showing signs of dieback while those at its southernmost end were visibly healthy at the commencement of this study. Transect 2 (located south of Transect 1) ran almost perpendicular to the face of the dieback front (Figure 2). Parkinsonia plants were identified, numbered, and mapped by running a 50 m tape measure and measuring distances at right angles along the tape to record each point in two dimensions. Each plant was then given an exact position using metres per degree.

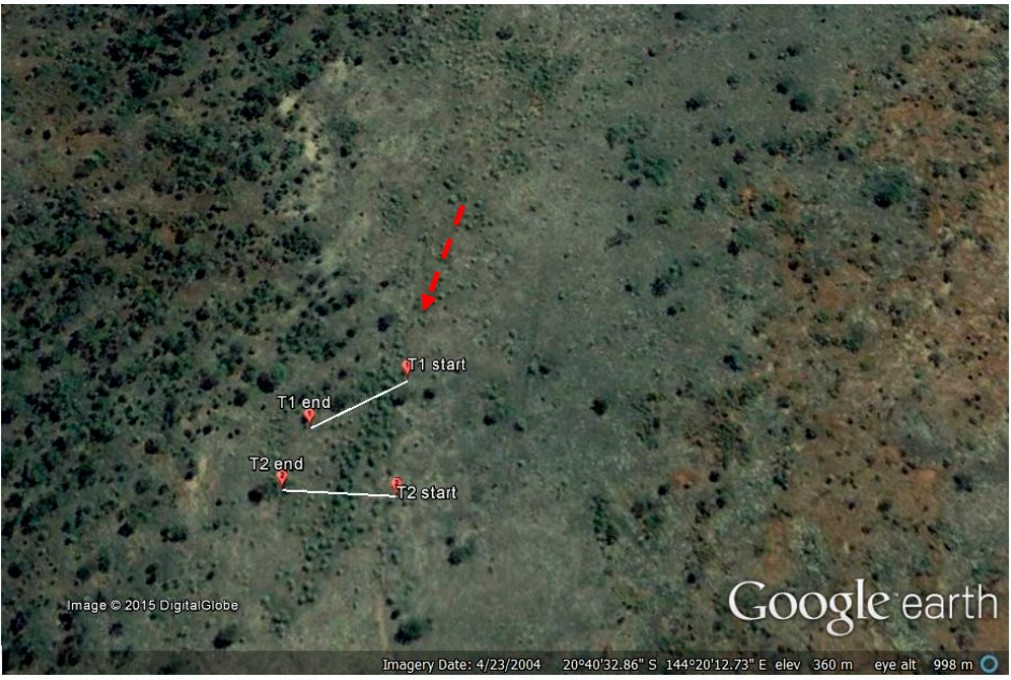

**Figure 2.** Transect orientation at Koon Kool Station. Transect lines are 50 m in length running in the general direction of east to west [15]. Red arrow shows the direction of travel of dieback front.

### 2.1. Plant Health Assessment

Individual plants were assessed for the presence or absence of dieback symptoms, living branch (a measurement of the amount of living branch material on the whole plant), and percentage of live foliage cover on live parts of the plant. Assessment of these parameters were recorded on a scale of 0–10 (where 0 = 0%, 10 = 100%). Measurements were taken at establishment in June 2005 and again in June 2006, September 2007, June 2008, and September 2012, a period spanning seven years and 3 months.

A final plant health rating scale was formulated to account for both relative live foliage cover and living branch. This gave an overall categorical health rating of 0–110 (0 indicating dead, and 110 indicating maximum vigour). This was calculated using the formula (live foliage cover + 1) × (living branch as a proportion of the whole crown) with results grouped into 4 health categories of 0 (dead); 1 (ratings 1–35); 2 (ratings 36–74); and 3 (ratings 75–110). The categorical method was chosen to use the combined data of living branch and live foliage cover as this provided the most realistic representation of plant health. The distance

of dieback movement was also observed in each assessment. The distance of dieback travel in Transect 1 was determined by calculating the furthest distance (m) of dieback movement (plant death) in relation to the closest dead plant from the year in question. Plants already exhibiting a rating of 0 from previous assessments were not included in calculations.

*2.2. Analysis of Plant Health and Movement of Dieback*

A Markov chain model (Microsoft Excel 2013) was created to predict the distribution of plant health groups within a parkinsonia population over a 25-year period. Plant vigour ratings from 2005–2008 were used to create this model, excluding the final assessment data of 2012 due to the time gap in assessments. This stochastic model was then used to predict plant mortality rates over 20 years.

*2.3. Isolation and Identification of Fungi Associated with Dieback-Affected Plants*

Parkinsonia stem samples (20 cm billets) were collected from dieback-affected plants during the establishment of the transects in June 2005 and returned to the laboratory to isolate and identify associated fungi [6]. Plant stems were cut into discs approximately 0.5 cm thick, avoiding material from the end of the segments to minimise the chance of surface contaminants. These discs were surface sterilised using 4% NaOCl for 3 min, rinsed in sterile water for 3 min, followed by a second rinse in sterile water for 1 min. The discs were then placed on 1/2 PDA (19.5 g/L Potato Dextrose Agar, Sigma-Aldrich, North Ryde, NSW, Australia) Petri plates using sterile forceps. These plates were placed in an incubator at 25 °C and observed daily. When fungal hyphae were observed growing from the stem discs, subcultures were made on fresh 1/2 PDA plates [6]. The process was repeated as necessary until pure cultures were obtained.

Cultures which did not readily produce spores were subcultured and exposed to black light with a 16 h day within the first four days of growth to encourage sporulation. In some instances where black light did not promote sporulation, water agar (20 g/L agar (Sigma-Aldrich, North Ryde, NSW, Australia) prepared with deionised water), was used with autoclaved parkinsonia seedlings (with cotyledons just emerged, seed casing removed) placed in the centre of the plate. Subcultures of the fungi were made onto the seedlings, and these were exposed to the same black light regime as described above.

Once cultures had produced spores, these were used to identify them to genus where possible. Initial identification of key isolates to species level was conducted by James Cunnington (Department of Primary Industries, Victoria, Australia) using morphological techniques.

**3. Results**

*3.1. Transects*

Satellite images (Figures 3 and 4) captured in 2004 and 2013 [15,16] and on ground photographs taken at the time of assessment demonstrate a collapse in the parkinsonia population in the transect study area (Figures 5 and 6). Individual plant health generally declined over consecutive years, with 98% of plants dying over the study period of 2005–2012 (Figures 7–10). In all cases, recovery was not observed once a plant had reached a rating of 0 (indicating that it was dead). This shows that although vigour may improve slightly in some plants from year to year, dieback was eventually lethal to all affected plants. In some cases, this increase in vigour reflected reshooting from the base of plants which were already in very poor health, subsequently, these secondary stems were found to be dead at the following assessment.

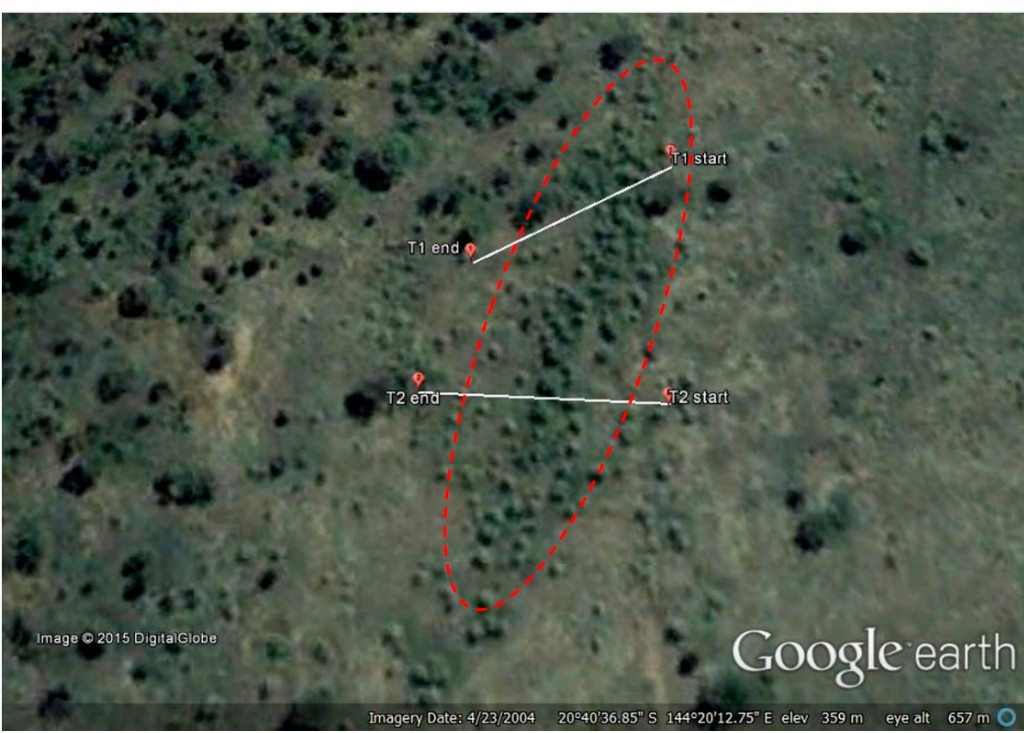

**Figure 3.** Satellite image of 50 m transects at Koon Kool Station one year before the trial establishment [15]. Red dashes outline a stand of parkinsonia plants.

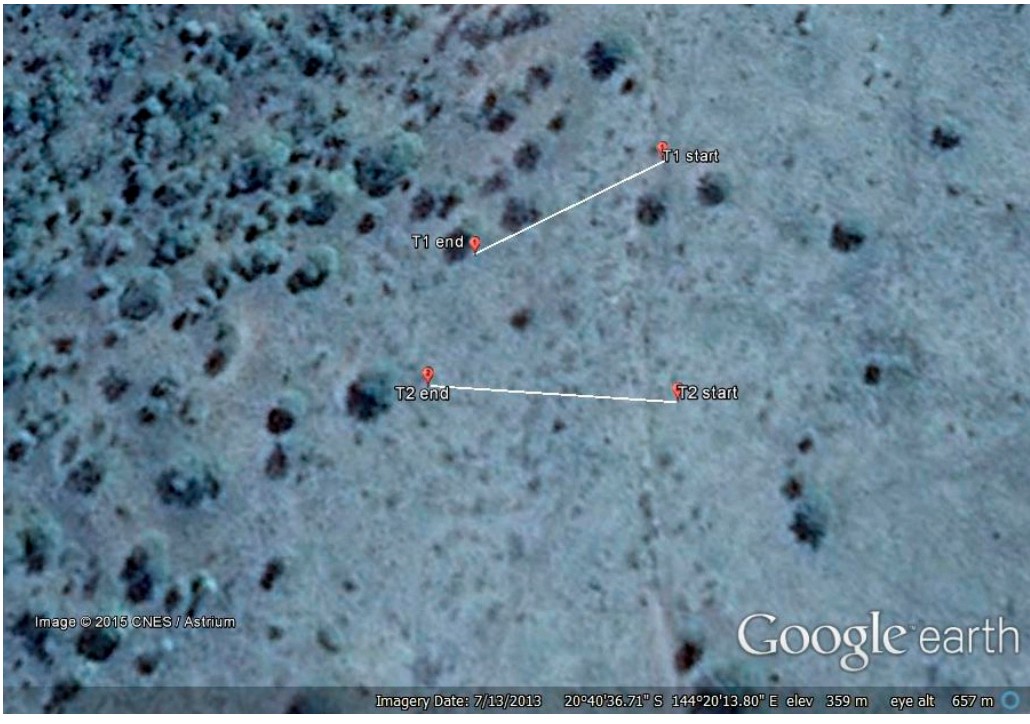

**Figure 4.** Satellite image of 50 m transects at Koon Kool Station one year after the final assessment (2013) [16]. Parkinsonia stand observed in 2004 no longer visible.

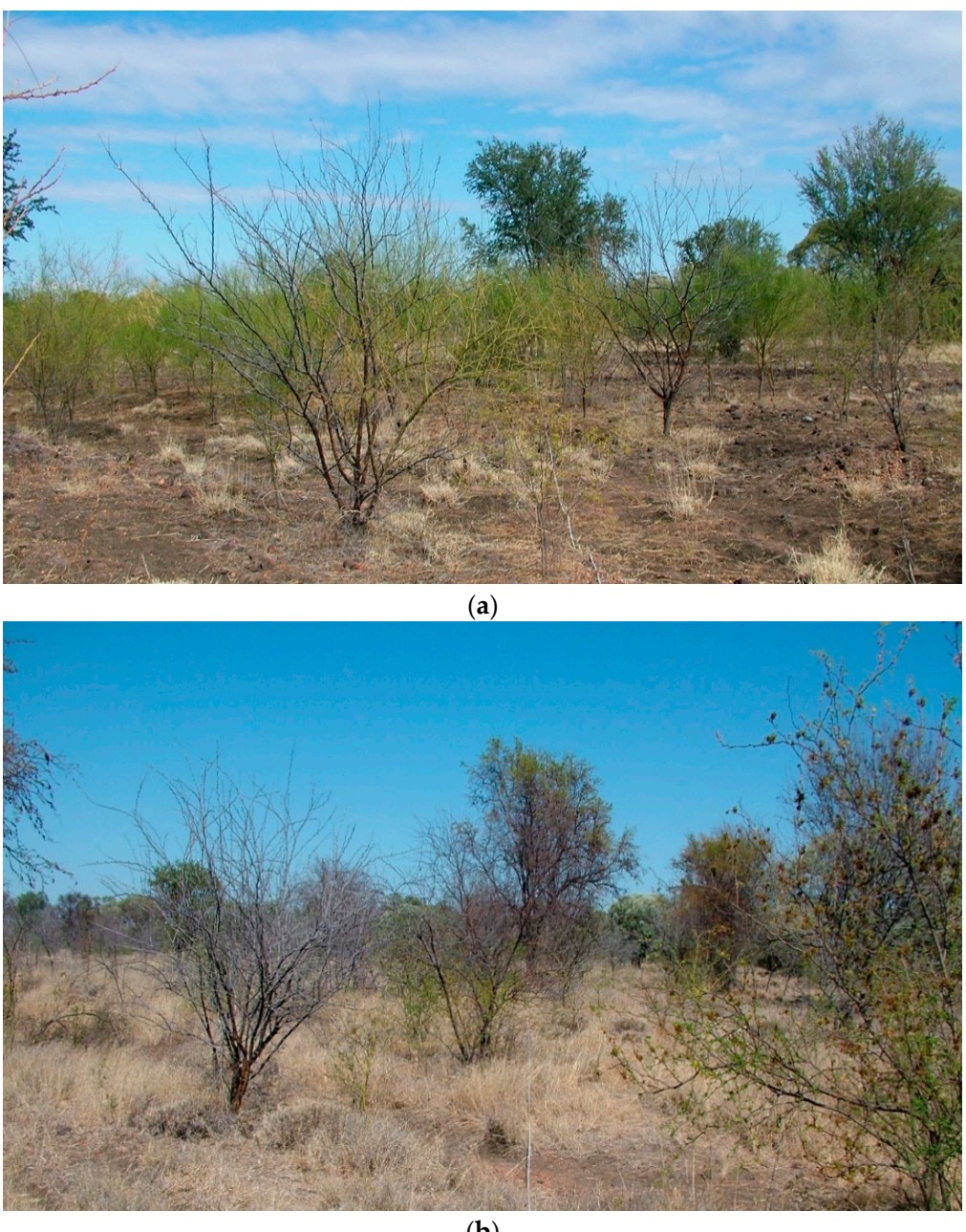

**Figure 5.** Transect 1 viewed at 0 m (start of transect) at establishment in 2005 (**a**), and final assessment in 2012 (**b**). Note the obvious disappearance of live parkinsonia plants (with light-green foliage) from the background in the second photograph.

The average distance of dieback movement was calculated at 7.7 m per year from the closest dead plant (ignoring dead plants from previous years) from 2005–2008 in Transect 1 (Table 1). Transect 2 was not used in this calculation due to the dieback moving in a perpendicular direction to the dieback front. (Figures 3, 8 and 10). Dieback observed in Transect 1 steadily progressed from the origin point over time. Movement of dieback along Transect 1 followed closely along the transect line over time, with dieback generally moving further from the origin point with each assessment. Seedling germination was observed every year, with new plants generally emerging where an adult plant had recently died. Most of the seedlings recorded died before the next assessment. There was a greater recruitment of seedlings in Transect 1 in the final assessment in 2012; however, it is likely that these did not persist, following the trend of previous years (Figures 7 and 9).

**Table 1.** Furthest distance (m) of dieback front movement (plant death) calculated from the closest dead plant (ignoring dead plants from previous years) in Transect 1 over a 7-year study. Trial established June 2005.

| Assessment | Transect 1 |
| --- | --- |
| June 2006 | 4.6 |
| September 2007 | 9.8 |
| June 2008 | 8.4 |
| September 2012 | 13.1 |
| Average 2006–2008 | 7.7 |

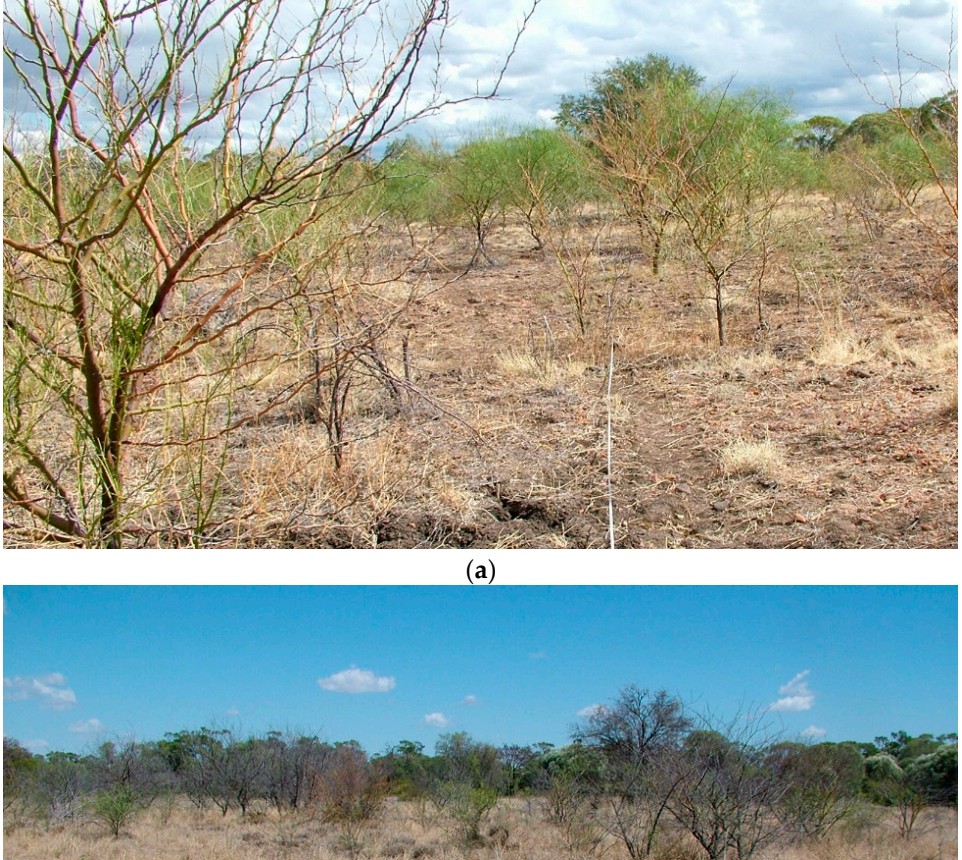

(**a**)

(**b**)

**Figure 6.** Transect 2 viewed at 0 m (start of transect) at establishment in 2005 (**a**); and final assessment in 2012 (**b**). Note the reduced parkinsonia population (light-green foliage) in the second photograph.

Using the Markov chain model, a steady decline of plant health is observed in a plant population. Commencing with a population displaying a 100% vigour rating at time 0 ($T_0$), the progression of reduced vigour is observed over 20 years. At the time point of 15 years ($T_{15}$), 99.5% of plants display a rating 0 (dead), with 100% mortality at $T_{20}$ (Figure 11). Using this model with 2008 actual values at $T_0$ and comparing 2012 predicted and actual values, a close match of percentages in each health category was seen, with slightly poorer health in the actual values demonstrated between the predicted and actual results (Figure 12).

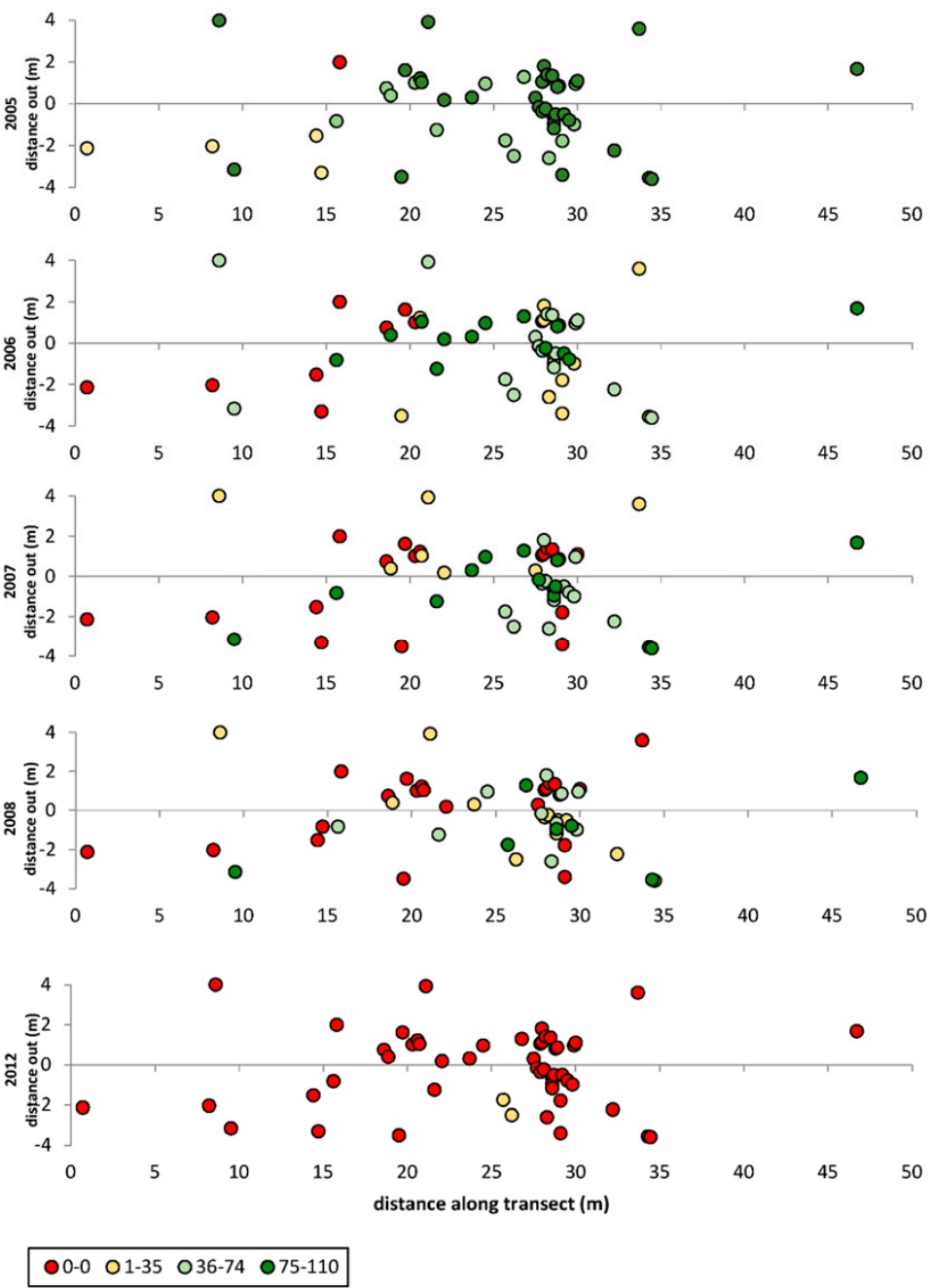

**Figure 7.** Transect 1, Individual Plant Health from June 2005 to June 2008 with a final rating in 2012. Health categories: ● = dead, ○ = ratings 1–35, ◐ = ratings 36–74, ● = ratings 75–110.

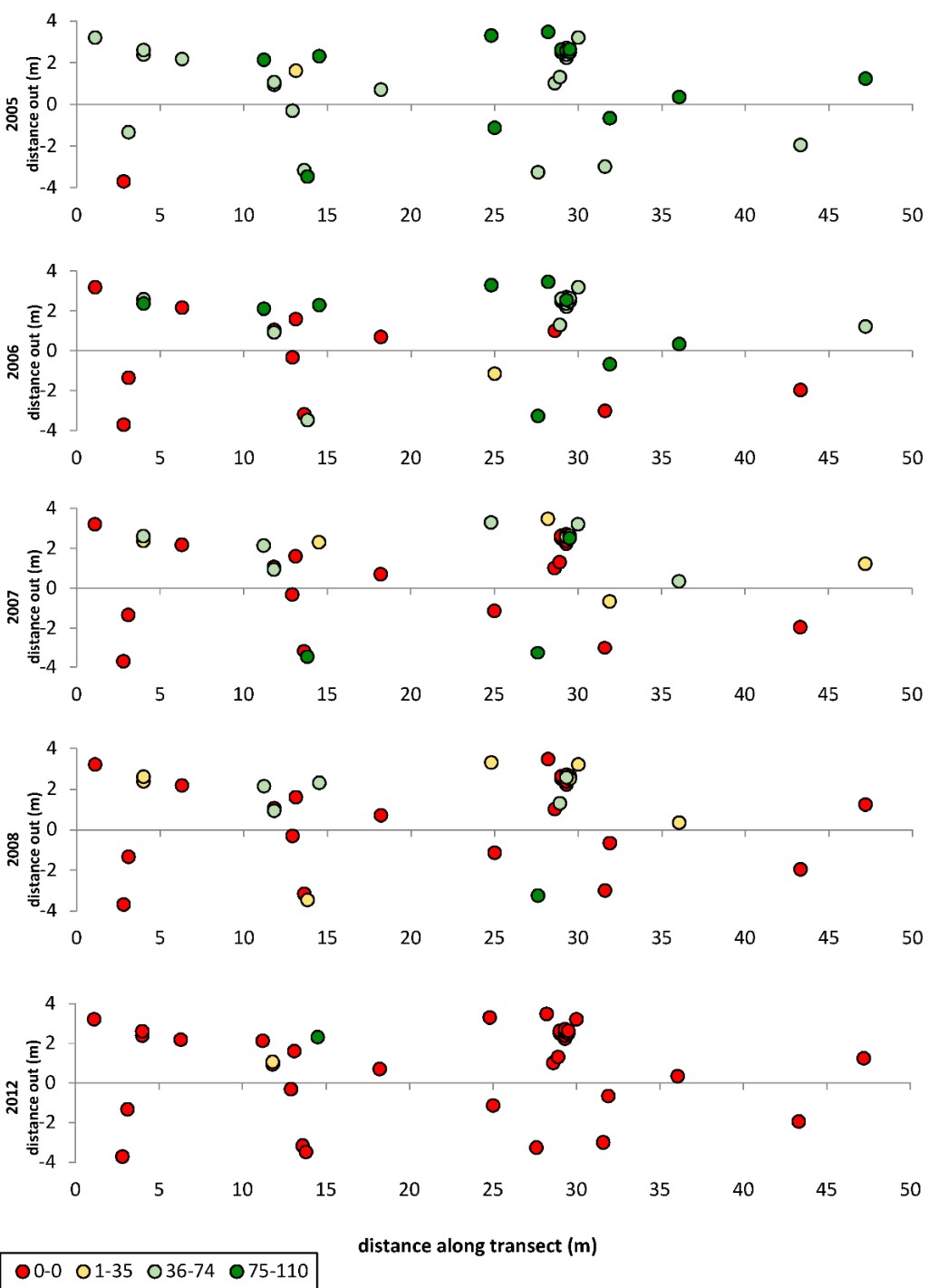

**Figure 8.** Transect 2, Individual Plant Health from June 2005 to June 2008 with a final rating in 2012. Health categories: ● = dead, ○ = ratings 1–35, ○ = ratings 36–74, ● = ratings 75–110.

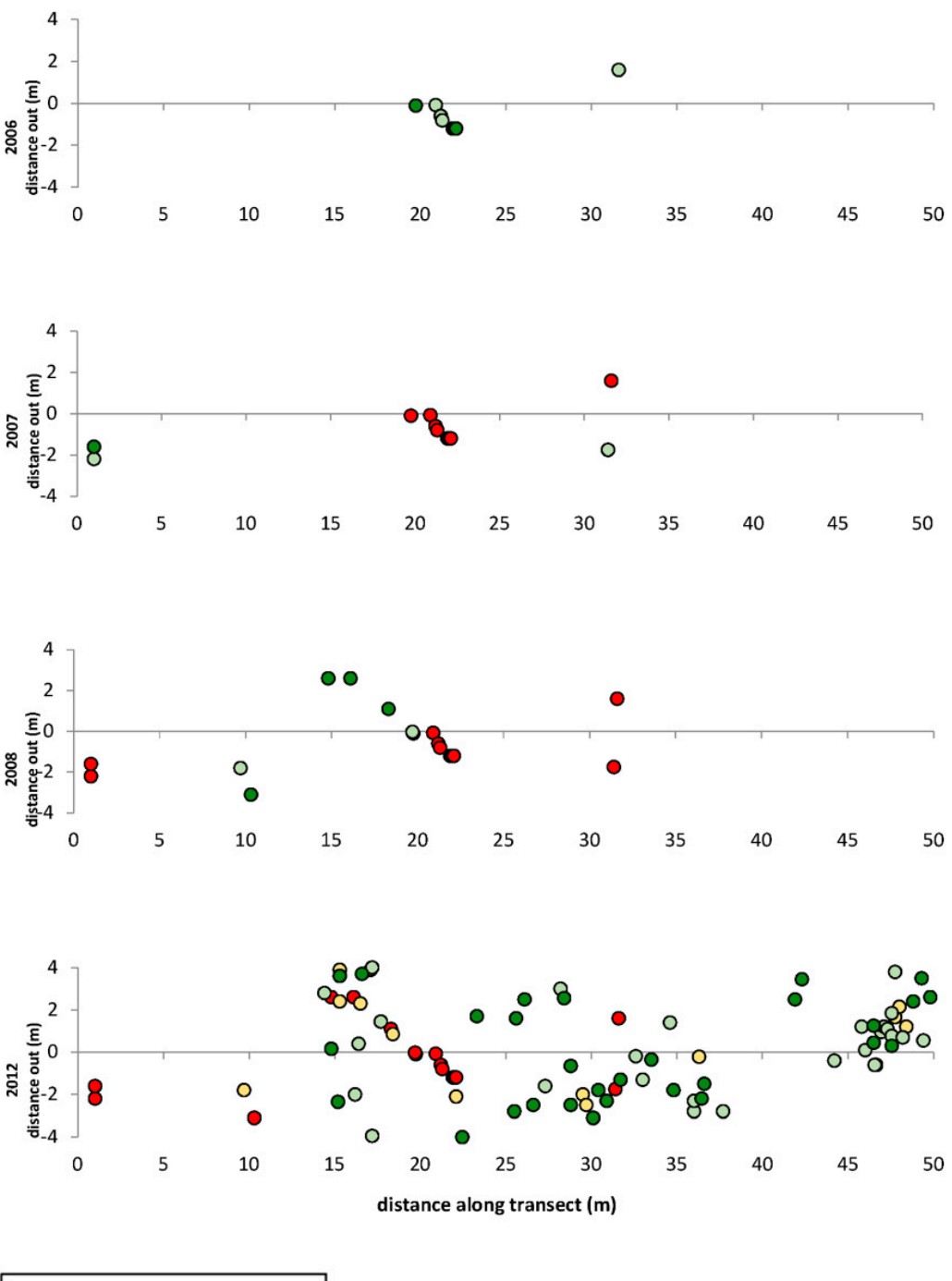

**Figure 9.** Transect 1, Individual seedling recruitment and health rating from June 2005 to June 2008 with a final rating in 2012. Health categories: ● = dead, ○ = ratings 1–35, ● = ratings 36–74, ● = ratings 75–110.

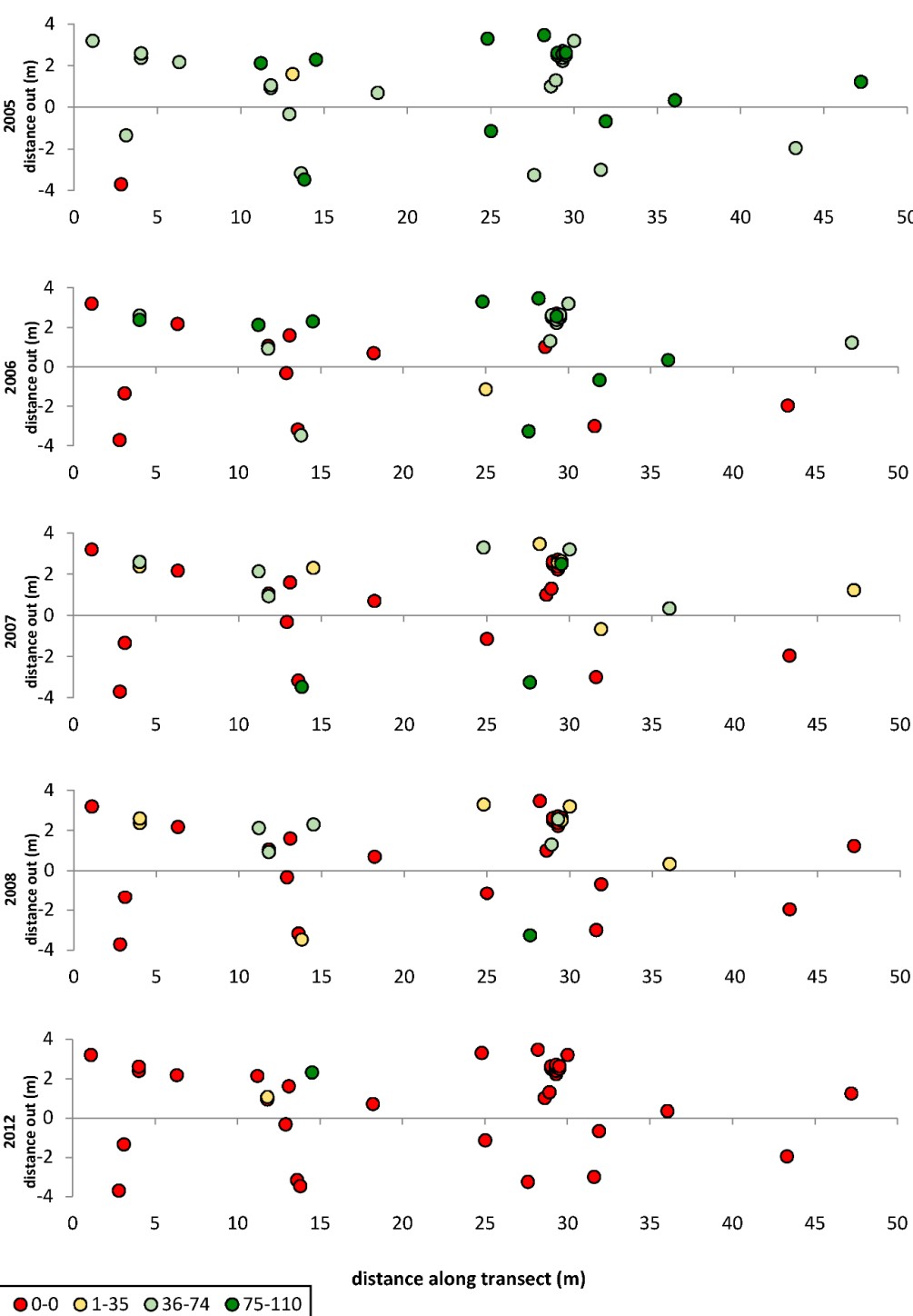

**Figure 10.** Transect 2, Individual seedling recruitment and health rating from June 2005 to June 2008 with a final rating in 2012. Health categories: ● = dead, ○ = ratings 1–35, ○ = ratings 36–74, ● = ratings 75–110.

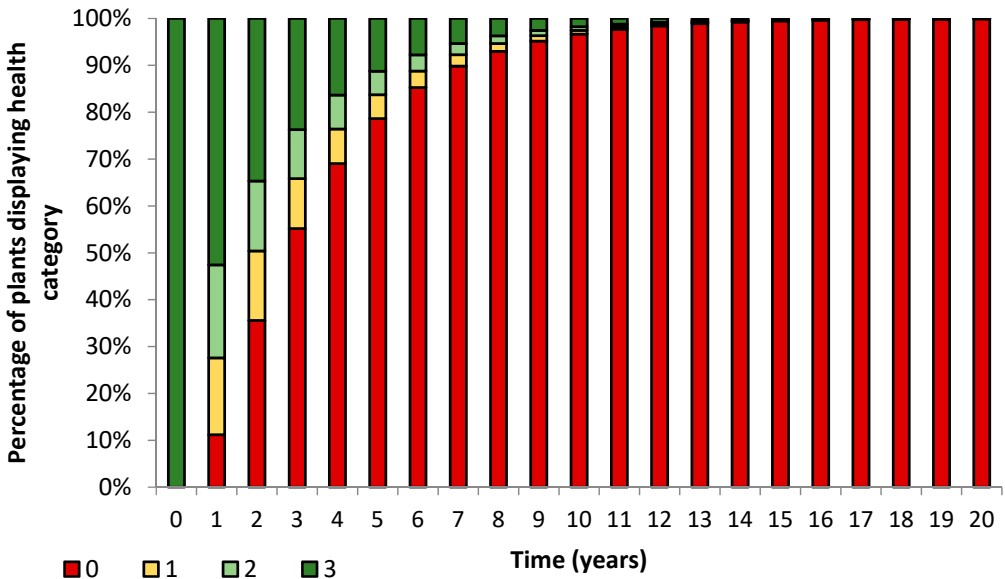

**Figure 11.** Predicted vigour ratings of dieback affected plants over a period of 20 years (0–25) using the Markov chain model. Health categories: 0 = dead (red), 1 = ratings 1–35 (yellow), 2 = ratings 36–74 (light green), 3 = ratings 75–100 (dark green).

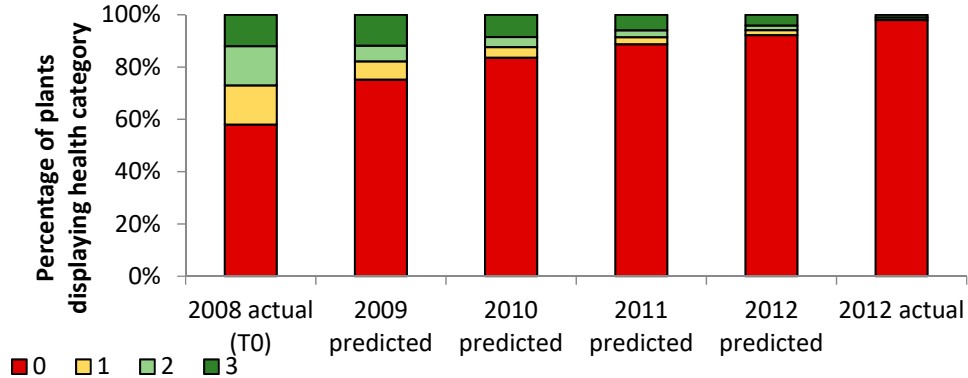

**Figure 12.** Percentage of predicted and actual health ratings of plants 2008–2012 using the Markov chain model. Health categories: 0 = dead (red), 1 = ratings 1–35 (yellow), 2 = ratings 36–74 (light green), 3 = ratings 75–100 (dark green).

*3.2. Isolation and Identification of Fungi Associated with Dieback-Affected Plants*

The presence of *Macrophomina phaseolina* was confirmed from samples taken from the transect site along with five other fungal species (Table 2).

**Table 2.** Identification of isolates retrieved from transect plants in 2005.

| Transect | Plant Number | Identification | Method of Identification | Date Collected |
|---|---|---|---|---|
| T1 | 9 | *Kabatiella bupleuri* | ITS sequence | 27 June 2005 |
| | 24 | *Bagnisiell examinans* | ITS sequence | 27 June 2005 |
| | 1 | *Aspergillus niger* | Morphology | 27 June 2005 |
| | 50 | *Aspergillus niger* | Morphology | 27 June 2005 |
| T2 | 9 | *Macrophomina phaseolina* | Morphology | 27 June 2005 |
| | 9 | *Phoma* sp. | Morphology | 27 June 2005 |
| | 36 | *Peyronellea curtsii* | Morphology | 27 June 2005 |
| | 19 | *Phoma* sp. | Morphology | 27 June 2005 |

## 4. Discussion

There are several theories on the phenomenon of dieback or decline and its interaction with the host plant. It can be problematic to assign a single cause to a dieback occurrence, as many factors contribute to decline [17,18]. Interactions between fungi and plants and their influence on the formation of a plant community may be complex with both biotic and abiotic factors swaying the outcome of these associations [10,11,19,20]. Plant pathogens play a key part in this process, shaping a landscape and creating change(s) in plant population dynamics. These changes may be expressed as the particular species present, the number of plants in a host population and genotypic range [9,19]. The impact of abiotic stress can incite weakly aggressive facultative pathogens and endophytic fungi to initiate disease [20]. This relationship between pathogens and plants and abiotic factors is multifaceted, with the movement, infection and symptom expression of these fungi being greatly influenced by the vegetation structure and spread; and the structure and pattern of plant species within the landscape being greatly influenced by the presence or absence of plant pathogens [19].

Isolates collected at the transect site included the presence of *Macrophomina phaseolina* [6]. *Macrophomina phaseolina* (Botryosphaeriaceae) was the only species with known pathogenicity on parkinsonia [21] found to be present in the transects at the commencement of the trial. Although this species was only found in 1/8 plants this could be attributed to a number of reasons. The sample size of affected plants may not have given an accurate representation of the fungi present in the area. Other species that were found are faster growing (e.g., *Aspergillus niger*) and may have outcompeted other fungi in the samples preventing detection of the dieback causal agent. The stage in dieback may also have been a factor contributing to the scarcity of potential causal agents found. Plants already affected by *M. phaseolina* (or any other undetected pathogenic fungi) may have since been invaded by saprophytes, resulting in isolations which no longer reflect the actual cause of the observed dieback. Given that *M. phaseolina* was the only known pathogen found in this study known to cause disease in parkinsonia [21], it will be the focus of this discussion.

Other dieback sites surveyed were commonly hosts to *Lasiodiplodia pseudotheobromae* and *Macrophomina phaseolina* [6], both being commonly associated with plant death [22–25], and found to have association with parkinsonia [6,26]. Both of these pathogens have also been found to be lethal to parkinsonia seedlings [21]. The composition of a plant community may be changed when plants are killed before reproducing. This may also be changed when seeds/seedling are also attacked by the pathogen/s responsible for killing the population of adult plants. In such instances, rapid decline in plant communities is observed [9]. Although pathogens are known to be responsible for much of this change and structure, this process remains largely understudied in naturally occurring environments [7]. Understanding the spatiotemporal spread of dieback movement through a stand of parkinsonia is vital in gaining an understanding of its effect on parkinsonia populations.

The movement of dieback through two transects over time demonstrated general decline in the health of individual plants, leading to almost complete death of the parkinsonia community. A host–pathogen encounter rate is used to partially explain the movement of disease through a stand of plants. This includes aspects of the environment such as the ever-changing number of hosts available for infection, different host species, pathogen reservoir, distance the plant pathogen (propagules) needs to travel between suitable hosts, the probability of transmission (depending on environmental requirements), resistant species and host vigour [9,19].

Infection by wet spore masses is somewhat limited to local infection of surrounding plants, which are likely to be the offspring of the original host plant, and share genetic similarities [27] creating a slow moving progression of disease moving through a group of plants. This trend of movement was observed in the study transects, with a disease front being observed to move at an average of 7.7 m through the stand over a three-year period. The variation in distance travelled over the transect may be due to many factors including inoculum movement through the stand (flooding/stock movement), or plant susceptibility due to external factors caused by environmental conditions. Flooding events

are generally observed in the summer months at Koon Kool Station, with the summer of 2008 to 2011 experiencing particularly high rainfall [13]. Plants which appeared healthy at the establishment of the transects were either suffering from dieback or dead within 3 years. Modelling of these data suggests that over 75% of the population in a stand of parkinsonia plants will be dead within 5 years once dieback has established at the site. Starting with a healthy population, this model suggests the number of plants displaying dieback will steadily increase until all plants are killed over a 20-year period.

The disease front observed moving through affected parkinsonia populations suggests that plants may be infected from direct contact with other infected plants, or at the least from infected plants bearing spores within close proximity, this likely to be facilitated through the movement of soil water movement. Parkinsonia generally grows in dense stands with plants in very close proximity to each other [28]. Root contact would occur in these instances, suggesting that infected plants could easily pass on infection to neighbouring plants through direct contact [29]. Root wounds are likely to be common in parkinsonia, with cracking clay at the transect site and often prominent in other areas of parkinsonia infestation (personal observation), resulting in minor root breakages providing an entry wound for dieback fungi [30].

Dispersal may also occur via conidia, spreading over a larger area [27,31], this may help account for movement within the region where seemingly 'random' parkinsonia plants exhibit signs of dieback (personal observation).

Despite some pathogens being widespread in occurrence, many are not likely to be uniform in distribution. As parkinsonia seeds are spread long distances by flood waters [28], it is possible they escape infection which would otherwise occur close to the parent plant in dieback-affected areas. Another significant agent of parkinsonia dispersal are cattle which have been known to eat the pods and disperse seeds [28], as well as carrying seeds attached to their coat in mud [2]. Circumstantial evidence which supports this theory are the observations of parkinsonia trees near loading yards where cattle from other properties are received and seeds from infested grazing areas are dislodged or deposited in dung.

Seed dispersal of *Platypodium elegans* has been investigated to determine the effect of distance and density of plants on infection by pathogens. It was found that the further from parent plants that seeds were dispersed, the less likely they were to succumb to death caused by pathogens [32]. Augspurger and Kelly [32] found that increased dispersal distance, decreased density and increased light levels and associated microclimates all reduced susceptibility to infection. This may also help to explain dieback occurrence in some regions, while stands of parkinsonia located on other properties within the same region appear healthy with no apparent signs of disease infection.

The occurrence of species can be controlled by naturally occurring plant pathogens [33], with re-generation being prevented if plants are killed before new generations are seeded. This is especially important in a weed such as parkinsonia where seeds are readily dispersed through water movement in times of flooding and may be carried to areas where dieback is not prevalent. Observations in naturally occurring dieback areas provide valuable information on the effectiveness of this phenomenon in controlling parkinsonia. Mature trees were observed, and so a high seed load could be expected in the soil in this area. Small numbers of seedling emergence were observed in dieback-affected areas, in comparison with healthy parkinsonia sites where dense blankets of seedlings are often seen to emerge (personal observation). This indicates that seeds are affected by dieback either as dormant seeds or after they have imbibed. In instances where germination was observed, recruitment of a new generation was not evident, with all seedlings dying after the emergence [21]. This understanding of the movement of dieback under field conditions supports the outcomes of a later study [34], where the inoculation of a large, healthy infestation of parkinsonia with bioherbicide capsules containing *Lasiodiplodia pseudotheobromae, Macrophomina phaseolina* and *Neoscytalidium novaehollandiae* resulted in effective management of this infestation where the average distance moved by the dieback front (68 m per year) was facilitated by the movement of ephemeral creek flows through the site.

## 5. Conclusions

The movement of a natural dieback through a population of an exotic, invasive woody weed in the Australian landscape was studied over a 7-year period. We found that the dieback front moved at a fast pace, at an average of 7.7 m per year from the closest dead plant. Once plants demonstrated signs of dieback, they did not recover.

A model predicting mortality from dieback suggests that 99.5% of plants would be dead within 15 years. When compared to the actual data from this site, the model slightly underestimated the actual mortality observed.

**Author Contributions:** Conceptualization, N.D.D. and V.J.G.; methodology, N.D.D. and V.J.G.; formal analysis, N.D.D.; investigation, N.D.D.; resources, V.J.G.; data curation, N.D.D.; writing—original draft preparation, N.D.D.; writing—review and editing, N.D.D. and V.J.G.; funding acquisition, V.J.G. All authors have read and agreed to the published version of the manuscript.

**Funding:** This research received no external funding; N.D. was supported by the University of Queensland postgraduate student scholarship.

**Acknowledgments:** We would like to thank Tony and Beth Kendall for hosting the trial. Shane Campbell and John McKenzie provided invaluable support in the field.

**Conflicts of Interest:** The authors declare no conflict of interest.

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
