# Peer review of "Temporal Movement of a Dieback Front in a Population of Parkinsonia in Northern Australia"

_agronomy, doi:10.3390/agronomy12020533_

Round 1

Reviewer 1 Report

Names of all fungal taxa throughout the text should be written in italics.

Introduction

I suggest adding a sentence describing the species Parkinsonia aculeata since it is not that known in Europe and some other parts of the world. For example, explaining in short that it is a woody species, considered as a weed in xxxxx plantations/or generally because of xxxxxxx. I suggest adding this sentence (or two) at the very beginning of the introduction.

When describing dieback symptoms, you should clearly state whether the dieback is affecting only the bark (phloem) tissue, or also the wood (xylem).

Materials and methods

Section beginning at the line 74 is more appropriate for the Introduction section, in Materials and methods you should shortly and precisely describe the monitoring methods you have used, and not comment on the reasons for choosing the particular area or repeat the aims of the study.

Two variables assessed as the indicators of plant health (living branch percentage and foliage cover percentage) should be more clearly explained and named at the beginning of the materials and methods section. It should be clearly stated what was measured, how and what were the exact names of these variables.

Results

Lines 163-169: These results should be presented more clearly. The exact number of emerged seedlings should be given, along with the percentages of survived ones. In Figures 9 and 10 these new emerged seedlings should be marked with different colours and not the same colours as used in the Figures 7 and 8.

Lines 170-173: The usage of Markov-Chain model should be stated and explained in the Materials and Methods section.

Discussion

Lines 241-245: For the first time in the manuscript you mention fungal isolates which might have caused the decline. These information should be stated in the Introduction section and then used in the discussion only to comment how these could have affected your results. Could the symptoms you have observed on the plants have been inflicted by the fungi you mention? Could the dieback have been caused by some abiotic factors or some other biotic factors? How can this be used in the future to manage the parkinsonia as the weed species?

The entire discussion and conclusions sections should be re-written. Currently, in the discussion section you are focusing on the association of dieback with pathogenic fungi, and you haven’t mentioned them anywhere earlier in the text and they were not a part of this research. Since your research was focused solely on the spatial and temporal development of dieback, you should focus on these results, and comment these results in the discussion. What are implications of your study for the future management of the parkinsonia?

For all other details, please refer to the pdf file with the comments (marked yellow).

Author Response

The authors thank Reviewer 1 for their careful and constructive comments which have improved the quality of this manuscript.

In response we have addressed the suggested improvements found in the edited PDF.  These can be viewed in the track changes in the returned manuscript file.

Introduction

I suggest adding a sentence describing the species Parkinsonia aculeata since it is not that known in Europe and some other parts of the world. For example, explaining in short that it is a woody species, considered as a weed in xxxxx plantations/or generally because of xxxxxxx. I suggest adding this sentence (or two) at the very beginning of the introduction.

This has been added to the manuscript

When describing dieback symptoms, you should clearly state whether the dieback is affecting only the bark (phloem) tissue, or also the wood (xylem).

This has been added to the manuscript

Materials and methods

Section beginning at the line 74 is more appropriate for the Introduction section, in Materials and methods you should shortly and precisely describe the monitoring methods you have used, and not comment on the reasons for choosing the particular area or repeat the aims of the study.

Two variables assessed as the indicators of plant health (living branch percentage and foliage cover percentage) should be more clearly explained and named at the beginning of the materials and methods section. It should be clearly stated what was measured, how and what were the exact names of these variables.

This has been clarified

Results

Lines 163-169: These results should be presented more clearly. The exact number of emerged seedlings should be given, along with the percentages of survived ones. In Figures 9 and 10 these new emerged seedlings should be marked with different colours and not the same colours as used in the Figures 7 and 8.

As the rating scale used to assess the seedlings was the same as the adult plants, the colours have been kept the same.  The legends and captions have been corrected.

Lines 170-173: The usage of Markov-Chain model should be stated and explained in the Materials and Methods section.

This has been clarified

Discussion

Lines 241-245: For the first time in the manuscript you mention fungal isolates which might have caused the decline. These information should be stated in the Introduction section and then used in the discussion only to comment how these could have affected your results. Could the symptoms you have observed on the plants have been inflicted by the fungi you mention? Could the dieback have been caused by some abiotic factors or some other biotic factors? How can this be used in the future to manage the parkinsonia as the weed species?

The entire discussion and conclusions sections should be re-written. Currently, in the discussion section you are focusing on the association of dieback with pathogenic fungi, and you haven’t mentioned them anywhere earlier in the text and they were not a part of this research. Since your research was focused solely on the spatial and temporal development of dieback, you should focus on these results, and comment these results in the discussion. What are implications of your study for the future management of the parkinsonia?

The method of collecting samples and pathogen identified at the site have been added to the methods and results.

Reviewer 2 Report

This is an interesting study, The movement of dieback through two transects over time demonstrated general decline in health of individual plants, leading to almost complete death of the parkinsonia community. A host-pathogen encounter rate is used to partially explain the movement of disease through a stand of plants.

The study goal is significant, not only in the local district but also worldwide and can be taken as reference for other plant/pathogens. Details are well described in method,

Some comments:

Fig.2 a and b and a seems are not in the same season, at least month, as referred to the dark green tree at the back row. The same to Fig.3. Taking consideration of vegetative growth characters in different phase maybe not convince.

Fig. 7 very interesting.

Table 1 it is confusing by saying ‘ignoring dead plants from previous years’. Does it mean the calculation is started from the last dead tree calculated from last year? Then it means the spread speed is increasing, 4.6 m (first year) around 9 m (second and third year), 13.1 m (fourth year), but the last line calculated an average speed, 2006-2008?

In other word, concerning the spread speed, will it be simply spread in an average speed or can be speed up (seems like in this data)? This part may be discussed.

Figure 11 the figure legends and the color rectangle are not fit. 1-3= ratings 1-30 (yellow) should be 1=

The same to Fig. 12. Labeled as Fig.4 should be correct.

Page l line 11, gain an understanding of the dynamics this disorder. Rephrase, hard to understand.

Author Response

The authors thank Reviewer 2 for their constructive input. In response to the comments refer below please:

Fig.2 a and b and a seems are not in the same season, at least month, as referred to the dark green tree at the back row. The same to Fig.3. Taking consideration of vegetative growth characters in different phase maybe not convince.

These figures have now been correctly numbered to Figures 5 and 6.  While the time of year may vary between these images, the key point is that parkinsonia is an evergreen tree which has green stems all year round.  Brown stemmed trees are the result of tree mortality.  The brown trees in the background demonstrate death by dieback.  

Fig. 7 very interesting.

Table 1 it is confusing by saying ‘ignoring dead plants from previous years’. Does it mean the calculation is started from the last dead tree calculated from last year? Then it means the spread speed is increasing, 4.6 m (first year) around 9 m (second and third year), 13.1 m (fourth year), but the last line calculated an average speed, 2006-2008?

In other word, concerning the spread speed, will it be simply spread in an average speed or can be speed up (seems like in this data)? This part may be discussed.

This is a reflection of the average spread of dieback along one transect from year to year. It is insufficient data to assume that the spread is speeding up over time, particularly given the reduction in spread in 2012. 

Figure 11 the figure legends and the color rectangle are not fit. 1-3= ratings 1-30 (yellow) should be 1=

The same to Fig. 12. Labeled as Fig.4 should be correct.

These have now been corrected

Page l line 11, gain an understanding of the dynamics this disorder. Rephrase, hard to understand.

 This has now been rephrased

Round 2

Reviewer 1 Report

In the Materials and Methods section, please precise the formula of a 1/2 PDA, i.e. the amount of agarose, potato dextrose, etc. Also, emphasise were the fungal isolates identified solely based on their morphological characteristics or were the molecular methods involved as well. If they were, you should describe them in detail.

In the results section, results regarding fungal isolates should be given with more details. How many isolates were obtained and how many of them were identified as M. phaseolina. Were there any other fungi identified? 

Author Response

The suggested amendements by reviewer 1 have been address.  Full details of the the microbioliogical media used in this study have been included.

More details about the fungi isolated from this trial site have been given and their identification approaches have been described.
